# COVID-19 Myocarditis: Prognostic Role of Bedside Speckle-Tracking Echocardiography and Association with Total Scar Burden

**DOI:** 10.3390/ijerph19105898

**Published:** 2022-05-12

**Authors:** Antonello D’Andrea, Luigi Cante, Stefano Palermi, Andreina Carbone, Federica Ilardi, Francesco Sabatella, Fabio Crescibene, Marco Di Maio, Francesco Giallauria, Giancarlo Messalli, Vincenzo Russo, Eduardo Bossone

**Affiliations:** 1Unit of Cardiology, Department of Traslational Medical Sciences, University of Campania “Luigi Vanvitelli”, Monaldi Hospital, 80131 Naples, Italy; luigicante3@gmail.com (L.C.); andr.carbone@gmail.com (A.C.); frasab93@gmail.com (F.S.); v.p.russo@libero.it (V.R.); 2Unit of Cardiology and Intensive Coronary Care, “Umberto I” Hospital, 84014 Nocera Inferiore, Italy; 3Department of Public Health, University of Naples Federico II, 80131 Naples, Italy; stefano.palermi@unina.it; 4Department of Translational Medical Sciences, University of Naples Federico II, 80131 Naples, Italy; fedeilardi@gmail.com (F.I.); francesco.giallauria@unina.it (F.G.); 5Unit of Cardiology, Scafati M. Scarlato COVID Hospital (ASL Salerno), 84018 Scafati, Italy; fabiocrescibene@gmail.com; 6Unit of Cardiology, Eboli Hospital (ASL Salerno), 84025 Eboli, Italy; marcodimaio88@gmail.com; 7Unit of Radiology, Sarno Hospital (ASL Salerno), 84087 Salerno, Italy; giancarlomessalli@hotmail.it; 8Cardiac Rehabilitation Unit, Cardarelli Hospital, 80131 Naples, Italy; ebossone@hotmail.com

**Keywords:** myocarditis, COVID-19, speckle-tracking echocardiography, cardiac magnetic resonance, total scar burden

## Abstract

SARS-CoV2 infection, responsible for the COVID-19 disease, can determine cardiac as well as respiratory injury. In COVID patients, viral myocarditis can represent an important cause of myocardial damage. Clinical presentation of myocarditis is heterogeneous. Furthermore, the full diagnostic algorithm can be hindered by logistical difficulties related to the transportation of COVID-19 patients in a critical condition to the radiology department. Our aim was to study longitudinal systolic cardiac function in patients with COVID-19-related myocarditis with echocardiography and to compare these findings with cardiac magnetic resonance (CMR) results. Patients with confirmed acute myocarditis and age- and gender-matched healthy controls were enrolled. Both patients with COVID-19-related myocarditis and healthy controls underwent standard transthoracic echocardiography and speckle-tracking analysis at the moment of admission and after 6 months of follow-up. The data of 55 patients with myocarditis (mean age 46.4 ± 15.3, 70% males) and 55 healthy subjects were analyzed. The myocarditis group showed a significantly reduced global longitudinal strain (GLS) and sub-epicardial strain, compared to the control (*p* < 0.001). We found a positive correlation (r = 0.65, *p* < 0.0001) between total scar burden (TSB) on CMR and LV GLS. After 6 months of follow-up, GLS showed marked improvements in myocarditis patients on optimal medical therapy (*p* < 0.01). Furthermore, we showed a strong association between baseline GLS, left ventricular ejection fraction (LVEF) and TSB with LVEF at 6 months of follow-up. After a multivariable linear regression analysis, baseline GLS, LVEF and TSB were independent predictors of a functional outcome at follow-up (*p* < 0.0001). Cardiac function and myocardial longitudinal deformation, assessed by echocardiography, are associated with TSB at CMR and have a predictive value of functional recovery in the follow-up.

## 1. Introduction

Coronavirus disease 2019 (COVID-19) was first noticed at the end of 2019 as several isolated cases of pneumonia in China [1,2]. In a few months, this disease rapidly progressed into a global pandemic with underlying hidden pathogenetic mechanisms increasing morbidity and mortality. One of the possible consequences of COVID-19 is represented by viral myocarditis; it appears with an infarct-like presentation consisting of typical symptomatology, and an increase of biomarkers and imaging features.

Cases of myocardial inflammation have been described in patients with a COVID-19 infection by a broad spectrum of clinical manifestations, ranging from paucisymptomatic forms, to predominantly arrhythmic manifestations, to cases of heart failure and cardiogenic shock [3,4,5]. Endomyocardial biopsy (EMB) is considered the gold standard tool for the diagnosis of myocarditis. However, EMB is an invasive exam and should be reserved only for life-threatening clinical presentations in which histological data can guide therapeutic choices, in accordance with ESC guidance for the diagnosis of cardiovascular diseases in COVID-19 patients [6]. This document also defines cardiac magnetic resonance (CMR) imaging, if available, as the primary diagnostic tool due to its high capability of tissue characterization. However, in some COVID-19 patients, the feasibility of diagnostic algorithm may be hindered by logistical difficulties related to the transport of patients in a critical condition into the radiology department.

Previous data on myocarditis in patients without COVID-19 infection have shown that echocardiographic parameters and especially speckle-tracking echocardiography (STE) correlate with scar burden, evidenced by CMR, with an important predictive value of the functional outcome.

The aim of our study was to compare STE parameters with scar distribution provided by CMR in order to evaluate the diagnostic role and prognostic stratification capacity of STE in the setting of COVID-19-related myocarditis, in which a bedside evaluation is certainly easier and more widely accessible.

## 2. Materials and Methods

### 2.1. Study Population

In this study, we prospectively included a selected sample of 55 consecutive patients admitted into Umberto I Hospital (Nocera Inferiore, Italy) and M. Scarlato COVID Hospital (Scafati, Italy) between August 2020 and March 2021 for COVID-19 disease, diagnosed with nasopharyngeal swab, for research of SARS-CoV2 nucleic acids with real-time PCR, and with a diagnosis of acute myocarditis confirmed by CMR.

Exclusion criteria were: age < 18 years old; severe clinical manifestation (sudden cardiac death, life-threatening arrhythmia, cardiogenic shock), and a high acoustic impedance of chest wall.

A total of 55 age- and gender-matched healthy subjects were enrolled in the control group. Control individuals were consecutively identified and enrolled if they were normotensive, had a normal 12-lead ECG exam and a preserved LV ejection fraction (>55%) and wall motion score index. Subjects were excluded if they had: (1) systemic hypertension at the moment of the visit (BP ≥ 135/85 mmHg as the average of three different repetitions) and/or were actually anti-hypertensive drug users; (2) known coronary disease; (3) any primary or secondary cardiomyopathy; (4) congenital heart disease; (5) at least mild mitral or aortic valvular insufficiency; (6) valvular stenosis; (7) any previous history of cardiac surgery or interventional procedure; (8) any kind of cardiac therapy; (9) previous history of cardioembolic stroke.

All patients released their written informed consent to participate to this study.

### 2.2. General Evaluation

All patients underwent medical history collection, physical examination and vital parameters measurement [heart rate (HR), blood pressure (BP) and body temperature (BT)]. Blood samples were picked up to evaluate inflammatory indices and enzymatic markers of myocardial necrosis.

### 2.3. Standard Transthoracic 2-Dimensional (2D) and Speckle-Tracking Echocardiography (STE) Analyses

The study protocol involved a complete 2D echocardiography and STE analysis, performed with a Vivid E80 Ultrasound Machine and an M5Sc transducer (GE Vingmed Ultrasound AS, Horten, Norway) at admission and after 6 months of optimal medical therapy (OMT). Echocardiographic images were analyzed offline, blinded to patient data, using a specific software (EchoPAC version 113, GE Vingmed Ultrasound AS).

Left ventricular (LV) cardiac diameters (end-diastolic and end-systolic) and thicknesses, LV systolic and diastolic function were assessed, according to American and European recommendations [7,8]. LV ejection fraction (LVEF) was calculated by Simpson’s biplane method [7], stroke volume (SV) with LV outflow tract (LVOT) diameter and the LVOT velocity—time integral was measured with a pulsed Doppler (PW). Cardiac output (CO) was determined by multiplying SV and HR.

For the STE analysis, the cine-loop clips were acquired from four-chamber, three-chamber and two-chamber apical views and analyzed offline. The peak systolic longitudinal strain was examined in all 17 LV segments and then the global longitudinal strain (GLS) was extrapolated from segmental values. The 2D strain of subendocardial, median wall and sub-epicardial regions of the myocardium were also calculated.

### 2.4. Cardiovascular Magnetic Resonance (CMR)

Cardiac magnetic resonance (CMR) was performed within 14 days of hospital admission with a 1.5 Tesla MRI Scanner (Aera XQ MRI, Siemens, Erlangen, Germany). 

The protocol of imaging was based on T2-weighted sequences to identify myocardial edema (area of reversible myocardial injury), on early gadolinium enhancement (EGE) in T1-weighted images to identify myocardial hyperemia (regional vasodilation is a peculiar aspect of tissue inflammation), and on late gadolinium enhancement (LGE) in T1-weighted images to identify necrosis or fibrosis (area of irreversible myocardial damage). The localization of LGE was represented through the 17-segment standard LV model [9]. The degree of LGE was defined by a 5-point scale, depending on the extension of hyperenhancement in the myocardial wall (0: absence; 1: 1–25%; 2: 26–50%; 3: 51–76%; 4: 76–100%).

The diagnosis of acute myocarditis was based on the latest Lake Louise consensus criteria [10].

### 2.5. Statistical Analysis

Descriptive statistics were performed: frequency and percentage were reported for the categorical variables; mean and standard deviation (SD) were used to summarize continuous variables.

Continuous variables were compared using paired and unpaired *t*-tests to study differences intra-group and between groups. To evaluate univariable relations, linear regression analyses and Pearson’s correlation test were used. Multivariable linear regression analysis was performed to investigate parameters independently associated with functional recovery at FU.

Parameters included in the multivariable analysis, with a stepwise backward elimination method, were: age, sex, BMI, mean blood pressure, LV diameters, LVEF, Doppler mitral inflow measurements, LA volume, strain measurements (LVGLS) and cardiac CMR indexes. The statistical significance was defined as two-sided *p* value < 0.01. To select optimal cut-off values of strain measurements, a receiver operating characteristic (ROC) curve analysis was evaluated. Intra-observer and inter-observer variability were evaluated with the coefficient of variation (COV), defined as the ratio of the standard deviation (σ) to the mean (μ) (%), and by Bland–Altman analysis. CV, 95% confidence intervals (CIs) and percent errors were reported. Statistical analyses were performed by SPSS for Windows release 21.0 (Chicago, IL, USA).

## 3. Results

Data from 55 patients with COVID-related acute myocarditis (mean age 46.4 ± 15.3 years) and 55 age- and gender-matched healthy controls (mean age 45.5 ± 16.7 years) were analyzed. Baseline clinical features and echocardiographic data of the overall population are reported in Table 1. The two groups were comparable for the main clinical variables, except for a significant increase in heart rate in the myocarditis group.

The myocarditis group showed LVEF, SV and CO were significantly reduced compared with control group (mean value 44.4 ± 5.7% vs. 54.4 ± 7.3%, *p* < 0.001). Furthermore, left atrial volume index, average E/e’ ratio, pulmonary artery systolic pressure and LV dimensions were significantly increased in the myocarditis group.

LVGLS was significantly impaired in patients with myocarditis compared to the control group (mean value −14.4 ± 5.2 vs. −22.1 ± 3.8%; *p* < 0.001) (Figure 1). Layer-specific strain values showed the same trend, with a main impairment of the epicardial and mid-wall layers.

All patients with myocarditis had myocardial edema at CMR, and myocardial hyperemia was found in 88.3% of patients (Table 2). LGE was detected in 49 of 55 patients (89.1%) and its distribution was linear and sub-epicardial in 87.3% and 70.6% of patients, respectively (Figure 1). TSB was 2.5 ± 1.3. Table 2 shows the CMR features of the myocarditis group.

A negative correlation between TSB and baseline LVEF (r = −0.4, *p* < 0.01) and a stronger positive correlation between TSB and baseline LVGLS (r = 0.65, *p* < 0.0001) were discovered at bivariate correlation analysis. Scatter plots of these correlations are respectively shown in Figure 2 and Figure 3.

LVEF, SV and LVGLS significantly increased after 6 months of OMT (Table 3).

Baseline LVEF (r = 0.52, *p* < 0.001), GLS (r = −0.46, *p* < 0.001) and TBS (r = −0.64, *p* < 0.0001) were correlated to LVEF at 6 months of FU (Table 4). Furthermore, baseline LVEF, LVGLS and TSB were independent predictors of functional recovery at 6 months, at multivariable linear regression analysis (LVEF β 0.35, *p* < 0.01; GLS β −0.36, *p* < 0.01; total scar burden β −0.53, *p* < 0.0001, Table 4).

Segmental peak systolic strain was significantly impaired in segments with LGE on CMR (−12.8 ± 3.5% vs. −18.4 ± 3.7%, *p* < 0.001; Figure 3) and a cutoff of −12% identified a segmental scar with a sensitivity of 78% and a specificity of 85% (AUC = 0.92; 95% CI 0.74–0.96; *p* < 0.001).

## 4. Discussion

### 4.1. Myocarditis in COVID-19 Patients

Cases of myocarditis are a widely described in COVID-19 patients. However, the prevalence of myocarditis in COVID-19 patients is difficult to define precisely; in fact, many reports often lack specific diagnostic criteria for the diagnosis of myocarditis and the rise of myocardial necrosis biomarkers could also be due to other causes (i.e., hypoxia, hypoperfusion, multiple organ failure, ventricular stress dysfunction) [11,12]. Overall, several studies report that myocardial damage occurs in 15–27.8% of severe COVID-19 pneumonia [13].

The exact pathogenetic mechanism of myocarditis in COVID-19 patients is not widely known, but direct viral damage or cell-mediated cytotoxicity are possible mechanisms hypothesized to be involved [14]. A proposed mechanism is based on the direct action of the virus, in particular the binding of the virus to ACE-2 receptors present on myocardiocytes. Although myocarditis has been clearly recognized by EMB, there is little evidence of myocarditis due to direct viral myocardial infection, but it may be due to a cytokine-induced inflammatory reaction. In fact, CD8 T lymphocytes migrate into the heart and cause a cytokine storm and then an inflammatory response (cell-mediated cytotoxicity).

The clinical presentation ranges from paucisymptomatic forms, to forms with acute coronary syndrome-like manifestations, to forms with a prevalent arrhythmic component and to cases of acute heart failure or cardiogenic shock. The highly variable clinical presentation and sometimes the difficulty related to the transport of patients in a critical condition to the radiology department make diagnosis challenging.

### 4.2. Diagnostic Tools in Acute Myocarditis and the Emerging Role of Speckle-Tracking Echocardiography

Although EMB is the diagnostic gold standard, it is indicated only in life-threatening conditions in which histological and immunohistochemical characterization (IHC) can influence therapeutic approach.

Conversely, in the setting of stable patients, CMR is the primary diagnostic exam for the diagnosis of acute myocarditis on the base of Lake Louise criteria (the presence of at least two criteria, including edema, hyperaemia and scar, predict myocardial inflammation with high diagnostic accuracy). The scar, identified with LGE, shows a distribution within the myocardial wall with a tendency to exclude the sub-endocardial region, giving us the possibility to distinguish this pattern from ischemia-related damage (sub-endocardial distribution). It has been widely demonstrated that CMR scar has a strong prognostic value, according to several studies that showed a correlation with the risk of major adverse cardiovascular events (MACE) and cardiovascular death [15,16].

Recently, several studies have underlined the role of STE in the diagnosis of myocarditis, demonstrating how the reduction of longitudinal strain correlates with areas of LGE at CMR assessment [17,18,19,20]. In particular, a previous study by our group already demonstrated that patients with acute myocarditis without COVID-19 infection had a reduction in LV longitudinal strain, which was most strongly decreased in the epicardial layer, with a strong correlation with scar burden detected with CMR. In this study we also found that GLS was an important prognostic factor for the functional recovery at 6 months of follow-up in optimal medical therapy. Therefore, longitudinal strain should be considered as a surrogate marker of fibrosis replacement, evaluated by LGE at CMR, with a strong value in prognostic assessment and risk stratification.

### 4.3. Main Results of This Manuscript

The results of this study confirmed the role of speckle-tracking echocardiography in predicting scar burden, detected by CMR, even in the setting of COVID-19-related myocarditis.

GLS in patients with COVID-19-related myocarditis appears to be impaired if compared to the control group or to the reference values described in the literature [21,22].

The reduction of longitudinal strain shows a layer-specific distribution. In particular, the tendency to preserve the sub-endocardial layer gives us the possibility to exclude ischemic injury. As a result, STE may also represent an important tool for differential diagnosis, especially in the case of myocarditis starting with an acute coronary syndrome-like presentation [23].

Of note, longitudinal strain correlates with scar in CMR more accurately than LVEF evaluated with the Simpson’s biplane method, confirming that STE has a greater sensitivity and is capable of detecting subclinical systolic dysfunction [24]. GLS impairment is directly proportional to total scar load, and segmental deformation can predict LGE areas with high diagnostic accuracy. While replacement fibrosis has clear prognostic value, STE, as an echocardiographic surrogate for LGE, has also shown a role in prognostic stratification.

Compared to CMR, echocardiography allows us to carry out a bedside diagnostic and prognostic assessment in order to overcome difficulties related to transport of COVID-19 patients who are in critical conditions.

### 4.4. Limitations

Our study has some limitations. First of all, the small sample size hinders the possibility to carry out precise assessments of the role of STE in COVID-19-related myocarditis.

Furthermore, there are intrinsic limitations of STE method. In particular STE depends on the frame rate. If the frame rate is too low, speckle tracking varies greatly from frame to frame, hampering precise characterization of myocardial deformation and systolic LV function. On the other hand, as the frame rate increases, there is a loss of spatial resolution.

Furthermore, the reduction of longitudinal strain may also be due to the increased wall stress in the acute setting since the strain is greatly influenced by hemodynamic conditions: preload and afterload. Finally, in the cases of mechanical ventilation, the acoustic window may not have been optimal to acquire images suitable for STE assessment. In fact, patients with high acoustic impedance of chest wall were excluded from the present study.

## 5. Conclusions

Previous studies identified STE as an important diagnostic tool in acute myocarditis. Our study confirmed that in the setting of COVID-19-related myocarditis, STE parameters are also surrogate markers of fibrosis replacement, evaluated by LGE at CMR. As a result, STE could play an important role in the prognostic assessment allowing a bedside risk stratification of COVID-19-related myocarditis.

## Figures and Tables

**Figure 1 ijerph-19-05898-f001:**
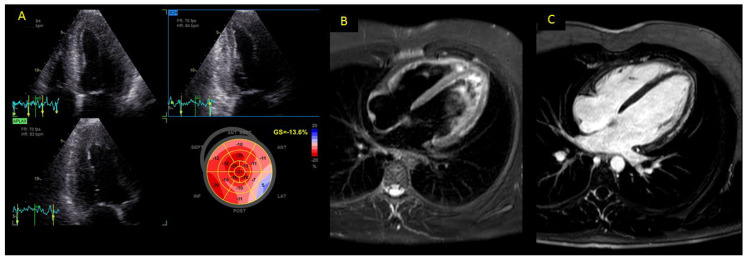
Panel (**A**) 17-segment bull’s-eye representation of LV strain in a patient with COVID-related myocarditis. Myocardial deformation (GLS—13%) was moderately impaired, especially in the lateral wall. Panels (**B**,**C**): cardiac MRI of the same patient, showing presence of myocardial edema in the stir T2-weighted sequence (Panel **B**) and of sub-epicardial scar tissue in lateral wall (Panel **C**) by LGE analysis.

**Figure 2 ijerph-19-05898-f002:**
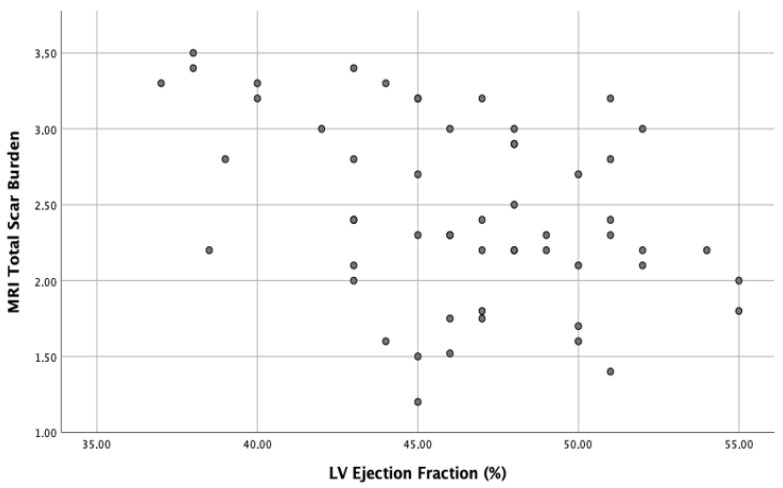
Scatter plot of negative correlation (r = −0.4, *p* < 0.01) between TSB (total scar burden) and baseline LVEF (left ventricular ejection fraction) in patients with myocarditis.

**Figure 3 ijerph-19-05898-f003:**
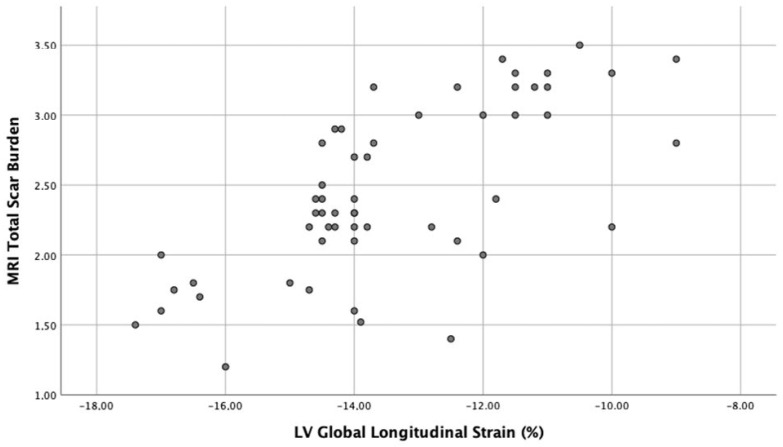
Scatter plot of positive correlation (r = 0.65, *p* < 0.0001) between TSB (total scar burden) and baseline LVGLS (left ventricular global longitudinal strain) in patients with myocarditis.

**Table 1 ijerph-19-05898-t001:** Baseline characteristics of sample size.

Category	Variables	Patients with Myocarditis (*n* = 55)	Healthy Controls (*n* = 55)	*p*-Value
**Clinical**	Age, years	46.4 ± 15.3	45.5 ± 16.7	>0.05
	Male gender (%)	40 (73%)	41 (74.5%)	>0.05
	HR, bpm	90.4 ± 19.4	74.6 ± 11.3	*
	SBP, mmHg	118.7 ± 17.3	123.3 ± 7.8	>0.05
	DBP, mmHg	78.4 ± 11.3	77.5 ± 4.9	>0.05
	BMI, kg/m^2^	27.5 ± 7.3	26.4 ± 4.8	>0.05
**Echocardiography**	IVSd, mm	9.5 ± 3.1	8.9 ± 3.3	>0.05
	PWd, mm	7.8 ± 3.7	7.4 ± 3.4	>0.05
	LVEDD, mm	53.4 ± 5.4	46.3 ± 3.7	*
	LVESD, mm	31.5 ± 6.3	22.5 ± 3.3	**
	LVMi, g/m^2^	50.7 ± 6.4	46.5 ± 3.3	>0.05
	LVEF, %	44.4 ± 5.7	54.4 ± 7.3	**
	SV, mL	52.4 ± 16.1	80.4 ± 15.6	**
	CO, L/min	3.7 ± 1.3	5.9 ± 2.1	*
	E/A ratio	1.8 ± 0.5	1.5 ± 2.5	**
	Average E/e’	13.9 ± 4.2	6.5 ± 3.5	**
	LAVi, mL/m²	33.4 ± 5.3	28.3 ± 4.2	*
	PASP, mmHg	41.4 ± 4.3	22.4 ± 2.4	**
**Speckle tracking**	GLS, %	−14.4 ± 5.2	−22.1 ± 3.8	**
	Epicardial GLS, %	−11.4 ± 5.8	−21.4 ± 3.1	***
	Mid-wall GLS, %	−13.9 ± 4.3	−22.4 ± 2.8	**
	Endocardial GLS, %	−15.9 ± 3.3	−23.7 ± 4.8	**

*: <0.05; **: <0.001; ***: <0.0001; HR: heart rate; SBP: systolic blood pressure; DBP: diastolic blood pressure; BMI: body mass index; IVSd: interventricular septum in diastole; PWd: posterior wall in diastole; LVEDD: left ventricular end diastolic diameter; LVESD: left ventricular end systolic diameter; LVMi: left ventricular mass index; LVEF: left ventricular ejection fraction (Simpson’s biplane); SV: stroke volume; CO: cardiac output; LAVi: left atrial volume index; PASP: pulmonary artery systolic pressure; GLS: global longitudinal strain.

**Table 2 ijerph-19-05898-t002:** Cardiac magnetic resonance features of patients with myocarditis.

CMR Features	*n* (%)
Edema	55 (100)
Hyperemia	48 (88.3)
LGE	49 (89.1)
LGE distribution	
-Linear	48 (87.3)
-Patchy	5 (9.1)
-Diffuse	2 (3.6)
LGE pattern	
-Epicardial	39 (70.6)
-Mid-wall	15 (27.6)
-Transmural	1 (1.8)
Pericardial effusion	13 (23.6)
TSB, mean ± SD	2.5 ± 1.3

CMR: cardiac magnetic resonance; LGE: late gadolinium enhancement; TSB: total scar burden; SD: standard deviation.

**Table 3 ijerph-19-05898-t003:** Echocardiographic features at baseline and at 6 months of follow-up in the population of patients with myocarditis.

	Baseline	6 Months FU	*p* Value
LVEF, %	44.4 ± 5.7	54.6 ± 4.1	*
SV, mL	52.4 ± 16.1	61.8 ±15.4	*
GLS, %	−14.4 ± 5.2	−16.8 ± 4.3	*
Epicardial GLS, %	−11.4 ± 5.8	−14.7 ± 5.3	*
Mid-wall GLS, %	−13.9 ± 4.3	−16.8 ± 7.3	*
Endocardial GLS, %	−15.9 ± 3.3	−18.9 ± 5.3	**
E/A ratio	1.8 ± 0.5	1.5 ± 0.3	*
Average E/e’ ratio	13.9 ± 4.2	8.8 ± 3.1	**
LAVi, mL/m²	33.4 ± 5.3	32.4 ± 5.2	>0.05
PASP, mmHg	41.4 ±4.3	31.3 ± 7.5	*

*: <0.05; **: <0.001; FU: follow-up; LVEF: left ventricular ejection fraction; GLS: global longitudinal strain; LAVi: left atrial volume index; PASP: pulmonary artery systolic pressure; SV: stroke volume; CO: cardiac output.

**Table 4 ijerph-19-05898-t004:** Univariable and multivariable analyses for functional outcome.

	Univariable Correlation Analysis			Multivariable Linear Regression Analysis		
	R	95% CI	*p* Value	Β	95% CI	*p* Value
Baseline LVEF	0.52	0.22; 0.72	**	0.34	0.23; 0.6	*
Baseline GLS	−0.46	−0.25; −0.62	**	−0.36	−0.28; −0.69	*
Baseline GLS epicardial	−0.55	−0.31; −0.57	***	−0.43	−0.38; −0.53	**
Baseline GLS mid-wall	−0.43	−0.34; −0.61	**	−0.32	−0.28; −0.64	*
Baseline GLS endocardial	−0.33	−0.15; −0.51	*	−0.28	−0.17; −0.55	>0.05
Total scar burden	−0.64	−0.51; −0.70	***	−0.53	−0.34; −0.64	***

*: <0.05; **: <0.001; ***: <0.0001; LVEF: left ventricular ejection fraction; GLS: global longitudinal strain.

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
