# Peer review of "COVID-19 Myocarditis: Prognostic Role of Bedside Speckle-Tracking Echocardiography and Association with Total Scar Burden"

_ijerph, 2022, doi:10.3390/ijerph19105898_

Round 1

Reviewer 1 Report

The study by D'Adrea et al is an interesting study correlating 

1) LV GLS with TSB

2) TBS and LV GLS with LVEF in 6-month follow up

 in patients with covid-19 and myocarditis

The study is well presented and the results are interesting for the reader adding depth to what is already known about the cardiovascular effects of covid-19

I have a couple of issues that in my opinion need to be addressed

1) How was the study population enrolled. Was the study retrospective or prospective. And if prospective how many patients were examined before the 55 patients were finally selected

2) How was the 6-month improvement on LVEF was defined (functional recovery). As a percentage of the initial LVEF, as a LVEF above a given value, as a nominal or as a continuous variable?

minor issues

I would prefer the p-values to be presented and not just the level of significant (just a personal preference)

Line 270 "Of note, longitudinal strain correlates with scar in CMR more accurately than FE evaluated with Simpson Biplane method"

do the authors mean LVEF instead of FE?

Author Response

Dear Reviewer 1:

We wish to thank you all for your constructive comments in this review. Your comments provided valuable insights to refine its contents and analysis. In this document, we try to address the issues raised as best as possible. Please find the requested modification through tracked-changes function of Microsoft Word

  • We improved grammar and syntax all over the manuscript
  • About sample size enrollement, we prospectively included a selected sample of 55 consecutive patients admitted in two italian hospital, who met our inclusion criteria: diagnosis of COVID-19 infection with PCR and diagnosis of acute myocarditis with CMR. As requested, we speicified it into these section
  • About the 6-months improvement of LVEF, as shown by Table 3, it is expressed as a percentage value: from 44.4% to 54.6% (therefore, almost a "normal value"): this highlights the optimal therapy efficacy against myocarditis
  • We absolutely agree with the preference of expressing p value numbers: however, the editorial rules foced us the change the style of tables and so we made this decision
  • Thanks for the typo of line 270: we changed it

Reviewer 2 Report

The authors aimed to assess longitudinal systolic cardiac function in patients with COVID-19 related myocarditis at echocardiography, and to compare these findings with cardiac magnetic resonance (CMR) results. They included 55 patients with acute COVID-related myocarditis and 55 (50? – please see below) controls. They concluded that cardiac function and myocardial longitudinal deformation, assessed by echocardiography, are associated with total scar burden at CMR and have a predictive value of functional recovery in the follow-up.

The study is well designed, written, and presented. The study is of practical clinical and scientific value.

However, there are some issues that need to be raised:

  1. Abstract: the sentence: “ Both patients with myocarditis COVID-19 related than healthy controls underwent standard transthoracic echocardiography…” is confusing. It should be rather: “ Both patients with COVID-19-related myocarditis AND healthy controls underwent standard transthoracic echocardiography…”
  2. In the abstract it is written that there were 50 healthy individuals included, while further in the text and in table 1 it is given the number 55. Please clarify.
  3. Paragraph 2.4: Why did the authors base the CMR diagnosis of acute myocarditis using old Lake Louise criteria instead of modified criteria (https://pubmed.ncbi.nlm.nih.gov/30545455/)
  4. Paragraph: 2.5: There is no information how testing for normality was performed.
  5. Table 1: The abbreviation CO (cardiac output) should be explained in the legend for table 1. The same applies to other abbreviations used in this table.
  6. In table 1, there are given LV dimensions/volumes and function based on echo study only. Since CMR study is the reference method for LV volumes and function, why didn’t the authors present CMR-derived data? Were there any differences between groups in CMR-based parameters such as LVEF, LVEDV, LVESV, LVM, LVSV and CO.
  7. Line 167: Since LVGLS has negative (below zero) values, I would prefer using the word “impaired” instead of “reduced”. The same applies to lanes 209, 263, and 272.
  8. Please add the information about correlation coefficient and p-value in the figures 2 and 3.
  9. Did the patients have follow-up CMR study?
  10. Line 224: there is a typo error. It is written “is base on”, it should be “is based on”
  11. Line 270. What does the abbreviation FE mean? Shouldn’t it be “EF” (ejection fraction)?
  12. References should be formatted according to the journal guidelines (e.g. Eur Heart J vs European Heart Journal)

Author Response

Dear Reviewer 2:

We wish to thank you all for your constructive comments in this review. Your comments provided valuable insights to refine its contents and analysis. In this document, we try to address the issues raised as best as possible. Please find the requested modification through tracked-changes function of Microsoft Word

  • We improved grammar and syntax all over the manuscript
  • We corrected several mistakes you shown us: we are very grateful for this
  • The healthy controls were 55 subjects, thanks for let us note!
  • We corrected the reference about Lake Louis criteria, adding the new modified version you suggested: it was a distraction error, since our CMR expert cardiologist confirmed us to have used them to diagnose myocarditis, so following latest guidelines
  • We improved tables and figures
  • Ablout LV values, we used echo parameters since these were more available and more easily "manageable", even if we recognize that CMR is the gold standard method. This was linked to availability of (bed-side) echocardiography at the moment of the enrollement
  • We did not use CMR data in the follow-up of this study: one of the reasons was to highlight the potential role of bed-side evaluation of patients, with the echocardiography as a very important tool
  • We corrected the reference style